# Plant Growth-Promoting Activities of Bacteria Isolated from an Anthropogenic Soil Located in Agrigento Province

**DOI:** 10.3390/microorganisms10112167

**Published:** 2022-10-31

**Authors:** Pietro Barbaccia, Raimondo Gaglio, Carmelo Dazzi, Claudia Miceli, Patrizia Bella, Giuseppe Lo Papa, Luca Settanni

**Affiliations:** 1Dipartimento Scienze Agrarie, Alimentari e Forestali, Università Degli Studi di Palermo, 90128 Palermo, Italy; 2Council for Agricultural Research and Economics, Plant Protection and Certification Centre, 90121 Palermo, Italy

**Keywords:** anthrosoils, cultivable bacteria, plant growth promoters, soil bacteria

## Abstract

Bacteria producers of plant growth-promoting (PGP) substances are responsible for the enhancement of plant development through several mechanisms. The purpose of the present work was to evaluate the PGP traits of 63 bacterial strains that were isolated from an anthropogenic soil, and obtained by modification of vertisols in the Sicily region (Italy) seven years after creation. The microorganisms were tested for the following PGP characteristics: indole acetic acid (IAA), NH_3_, HCN and siderophore production, 1-aminocyclopropane-1-carboxylate deaminase activity (ACC) and phosphate solubilization. The results of principal component analysis (PCA) showed that *Bacillus tequilensis* SI 319, *Brevibacterium frigoritolerans* SI 433, *Pseudomonas lini* SI 287 and *Pseudomonas frederiksbergensis* SI 307 expressed high levels of IAA and production of ACC deaminase enzyme, while for the rest of traits analyzed the best performances were registered with *Pseudomonas* genus, in particular for the strains *Pseudomonas atacamensis* SI 443, *Pseudomonas reinekei* SI 441 and *Pseudomonas granadensis* SI 422 and SI 450. The in vitro screening provided enough evidence for future in vivo growth promotion tests of these eight strains.

## 1. Introduction

Unlike soils created by natural processes, anthropogenic soils (anthrosoils) have been affected, altered, or created by human activity. These soil types are generally found in different continents, and they are typically divided into four categories: urban, agricultural, mine related and archaeological soils [1]. The dumping of various materials for agricultural uses brings the soil to time zero, from the pedogenetic perspective; indeed, such events are seen as catastrophic [2]. In southern Italy, a lot of pedotechniques are used to improve the economic value of soils; these types of soil management are used in some areas of the Sicily region (Italy). Often, the original soils are covered with marly limestone, and subsequently plowed in order to improve the suitability of the areas for table grape cultivation [3]. Changes in the chemical composition of soils influence biological activities [4]. Thus, bacterial communities are subjected to new equilibriums that can affect plant growth.

Plant growth-promoting bacteria (PGPB) represent a huge and heterogenous group of bacteria that can be found as free-living in bulk soil or in rhizosphere, interacting in a mutualistic relationship with a huge variety of plant species [5,6]. They are involved to varying extents in the improvement of plant growth through several mechanisms [7]. Furthermore, PGPB are able to colonize all types of natural environments; in studies carried out by Antoun and Kloepper [8], around 5% of the root microflora is composed by PGPB. In particular, rhizobacteria possess different modes of action; these mechanisms are split directly and indirectly and provoke the improvement of plant physiology and the defense against phytopathogens [9].

PGPB can act as biofertilizers; in fact, they can increase plant growth thanks to the solubilization of some elements (mainly P, K and Zn), nitrogen fixation and production of siderophores (small molecules able to improve iron uptake capacity) [10,11,12,13]. PGPB also influence plant growth through the production of a series of organic substances, namely phytostimulators or plant growth regulators [14]. These compounds include the most important plant hormones: indole acetic acid (IAA), cytokinins, gibberellins and ethylene enzyme suppressors [15,16,17,18]. Furthermore, some of these bacteria might be considered valid alternatives to common pesticides; indeed, they have the capacity to produce antibiotics, HCN and hydrolytic enzymes that directly contrast the phytopathologies, but they are also able to compete with a high number of plant pathogens through indirect methods consisting in the search for radical exudates competing with the pathogens in order to obtain the nutrients [19,20]. PGPB are commonly applied in bioremediation strategies in order to remove or immobilize soil pollutants such as herbicides, pesticides, solvents, organic compounds and heavy metals [21]. Finally, these microorganisms can be employed to help plants overcome stresses of a biotic and abiotic nature [22,23,24].

The characterization of soil bacteria for their useful contributions in stimulating plant growth is of paramount importance in evaluating the positive traits of the natural microbial communities of soils subject to human modification. To this end, the PGP aptitude of several bacterial strains, detected at dominating levels from an anthropogenic soil of the Sicily region, were tested with the main aim of determining the state of health of the soils from a microbiological point of view. All strains were screened for IAA, NH_3_, hydrogen cyanide, 1-aminocyclopropane-1-carboxylate deaminase (ACC), siderophore production and phosphate solubilization in order to estimate the positive functional role of the native bacterial community of these soils in supporting plant growth.

## 2. Materials and Methods

### 2.1. Plant Growth-Promoting Ability Assays

Bacteria used for the following assays were isolated and genetically characterized from an anthropogenic soil modified from Typic Haploxererts, located in the district of Giordano area within the Palma di Montechiaro (Agrigento, Italy) countryside, which is characterized by a Mediterranean climate [25].

The quantification of IAA was performed applying the method of Wholer [26]. The IAA was used for the construction of a standard curve in a range between 0 and 20 mg/L of water, using the Jenway Ltd. model 6400 (Dunmow, UK) spectrophotometer at 535 nm. All bacterial strains, previously stored at −80 °C, were cultured in nutrient broth overnight (Oxoid, Milan, Italy); afterwards, by centrifuging the culture media at 7000 rpm for five minutes, the cells were recollected and then cultured for 24 h at 37 °C in 3 mL of phosphate buffer (pH 7.5) containing 1% (*w*/*v*) of glucose and tryptophan. After incubation, cell suspensions were transferred into a solution of 2 mL of 5% (*v*/*v*) trichloroacetic acid and 1 mL of 0.5 M CaCl_2_. The solution was filtered through Whatman No. 2 filters (Whatman International Ltd., Maidstone, UK) and 3 mL of filtrate was added along with 2 mL of Salper solution (2 mL 0.5 M FeCl_3_ and 98 mL 35% (*v*/*v*) perchloric acid). The absorbance was measured at 535 nm after an incubation of half an hour in the dark at 25 °C.

In order to assess the ability to generate NH_3_, all bacteria were grown in peptone water for 72 h at 30 °C. Nessler’s reagent was added to each tube (0.5 mL), and the test was considered positive for NH_3_ production if broth color turned yellow-brown [27].

Hydrogen cyanide production was tested in Petri dishes using a modified nutrient agar (4.4 g/L of glycine). A filter paper Whatman no. 1 (Whatman International Ldt) was dipped in a solution prepared with 2% (*w*/*v*) sodium carbonate and 0.5% (*v*/*v*) picric acid, and was laid onto the surface of the agar medium; 10 µL of microbial solution was taken by a refresh tube with a concentration of 10^9^ CFU/mL, and was spread in a petri dish with the modified media and cultivated at 30 °C for 4 d. After that, colonies that acquired an orange or red color were considered positive for HCN production [5].

The synthesis of siderophores was carried out on a Chrome azurol S agar medium. Bacterial spots were transferred directly from growth plates, and after 48 h of incubation at 30 °C, the appearance of a bright orange halo surrounding the colonies indicated that siderophores had been produced in each single strain [28].

The method of Honma and Shimura [29] was modified to determine ACC-deaminase activity of bacterial strains. Pellets of bacterial cells were obtained as reported above. The cells were then resuspended in 5 mL of 0.1 mol/L Tris-HCL at pH 7.6, and after centrifugation at 16,000 rpm for 5 min, the pellets were further resuspended in 2 mL of 0.1 mol/L Tris-HCl at pH 8.5. The cell suspensions were added with 30 μL of toluene and vortexed for 30 s, and 200 μL of each cell suspension was transferred into a microtube, adding 20 μL of 0.5 mol/L ACC, and incubated at 30 °C for 15 min. After that, 1 mL of 0.56 mol/L HCl was added into a microtube that had been previously incubated, and the mix was homogenized and centrifuged for 5 min at 16,000 rpm at room temperature; 1 mL of supernatant was taken and mixed with 800 μL of 0.56 mol/L HCL, and 2 mL of 2,4-dinitrophenylhydrazine reagent was added to the mixture, vortexed and incubated at 30 °C for 30 min. Finally, 2 mL of 2 mol/L NaOH was added to the solution, and the absorbance was read spectrophotometrically at 540 nm. The measurement of α-ketobutyrate after hydrolysis of ACC is the basis of this method. The values obtained by this protocol was used to estimate the amount of μmol of α-ketobutyrate produced by the tested strains; these values were compared to a standard curve, obtained by adding 2 mL of 2,4-dini- trophenylhydrazine to each standard in a range between 0.1 and 1 μmol of α- ketobutyrate; the solution was vortexed and cultured at 30 °C for half an hour. The absorbance of the solution was measured at 540 nm after the addition of 2 mL 2 mol/L NaOH.

Phosphate solubilization was tested on a Pikovskaya medium (PVK). This medium had the following composition: 10 g/L glucose; 5 g/L Ca3(PO_4_)2; 0.5 g/L (NH_4_)2SO_4_; 0.2 g/L NaCl; 0.1 g/L MgSO_4_·7 H_2_O; 0.2 g/L KCl; 0.5 g/L yeast extract; 0.5 g/L MnSO_4_·H_2_O; and 0.002 g/L FeSO_4_·7 H_2_O [30]. After 15 d, the width of the halo around the colonies was measured, and colony diameter was subtracted from the total diameter. The phosphate dissolution rate was calculated using the formula (size of colony + size of clear zone)/diameter of colony.

Unless otherwise indicated, all chemicals and reagents were purchased from Sigma-Aldrich (Milan, Italy). 

### 2.2. Statistical Analysis

IAA and ACC data from the screening of microorganisms were analyzed using the One-Way Variance Analysis (ANOVA). Version 7.5.2 of the XLStat software for Excel was used for the analysis (Addinsoft, New York, NY, USA). The various strains put through the tests were compared using Tukey’s test. P values below 0.05 were considered statistically significant and are denoted by different letters.

To evaluate the correlation between the microorganisms and the parameters measured with the tests, the principal component analysis (PCA) was used. The number of major factors with eigen values greater than 1.00 were chosen using the Kaiser criterion [31]. The statistical significance within the dataset was examined with Barlett’s sphericity test [32].

## 3. Results

### 3.1. Plant Growth-Promoting Ability Assays

Results obtained from the PGPB screening are reported in Table 1. Statistical treatment of results of IAA production generated 20 different groups. The three largest genera of bacteria analyzed (*Brevibacterium*, *Bacillus* and *Pseudomonas*) showed a capacity to produce highly variable IAA. In particular, the *Brevibacterium* group included strains with no ability to generate IAA (strain SI 325) and strains with a high IAA production, until 7.37 mg L^−1^ was registered for *Brevibacterium frigoritolerans* SI 433. A similar variability was observed for the *Bacillus* genus with *Bacillus halotolerans* strain SI 339 unable to express this character until 6.94 mg L^−1^ was displayed by *Bacillus megaterium* SI 404 and *Bacillus cabrialesii* SI 428. Regarding *Pseudomonas*, which included 14 different bacterial strains, the range recorded was narrower than those of the two previous genera, from 0 mg L^−1^ of *Pseudomonas granadiensis* SI 450 to 6.50 mg L^−1^ of *Pseudomonas reinekei* SI 441. In addition to these three genera, there are interesting bacteria belonging to other genera such as *Streptomyces, Micrococcus, Sinorhizobium* and *Stenotrophomonas*, which possess the ability to produce consistent amounts of IAA.

Only 21 strains resulted positive for the NH_3_ production assay by turning the medium color to yellow-brown (Figure 1A). Regarding the major taxonomic bacterial groups, only three strains of *Br. frigoritolerans* (SI 264, SI 312, and SI 400) and three strains of *Bacillus* (*B. megaterium* SI 408, *B. halotolerans* SI 339 and *B. cabrialesi* SI 428) resulted positive for this test, while nine *Pseudomonas* strains, belonging to five different species, generated NH_3_. This character also registered positive for *Lysobacter soli*.

Only eight strains among the totality of the screened bacteria resulted positive in the HCN test by turning the filter paper color from yellow to orange-red (Figure 1B). All these bacteria were *Pseudomonas*. In particular, the species able to generate HCN were: *Pseudonomas brassicacearum, Pseudomonas frederiksbergensis, Ps. reinekei, Pseudomonas atacamensis, Ps. granadensis* and *Pseudomonas lini*.

A higher percentage of strains resulted positive for siderophore production. Thirty-one strains, including all *Pseudomonas* of the collection, produced siderophores. Among the strains belonging to other taxonomic groups, this capacity was shown by four *Br. frigoritolerans* strains and five *Bacillus* belonging to the species *B. megaterium*, *Bacillus tequilensis* and *Bacillus halotolerans.*

The results of the ACC test are reported in Table 1. Statistical analysis showed a great variability of data, indicating 28 different groups. The strain that showed the highest production of α-ketobutyrate was *Br. frigoritolerans* SI 433 with 80.58 nmol. *Bacillus* genus showed a high percentage of positive strains, with 9 out of 14 strains producing α-ketobutyrate after the hydrolysis of ACC. Furthermore, the *Pseudomonas* genus displayed a high percentage of positive strains in this test, with 11 out of 14 strains tested. Within the *Pseudomonas* group, all strains that tested negative belonged to the species *Pseudomonas lini*, even though the strain showing the highest α-ketobutyrate production is *Ps. lini* SI 287, with 62.28 nmol. Furthermore, all strains of *Peribacillus* displayed production of this acid.

Results highlighted that all bacteria able to solubilize phosphate belonged to the *Pseudomonas* genus, and the biggest halo diameter (8 mm) for phosphate solubilization was recorded for *Ps. lini* SI 270 (Figure 1C). The other three strains showing this character were *Streptomyces silaceus* SI 332, *Sinorizhobium melitoti* SI 240 and *Variovorax paradoxus* SI 435.

### 3.2. Statistical Analysis

The correlation of the tested bacteria with PGP abilities, evaluated by production of IAA, siderophore, hydrogen cyanide, NH_3_, expression of ACC deaminase activity and solubilization of phosphate, was analysed by PCA (Figure 2). The results highlighted how, for the first 2 components (PC1 and PC2), the eigen value reached 2.03 and 1.10, respectively. The 33.83% of total variability was expressed by the first component, while PC2 accounted for the 18.38%; thus, PC1 and PC2 together accounted for 52.22% of total variability. The graphical biplot shows that the first component (F1) had a strong influence on IAA and production of the ACC deaminase enzyme, while PC2 showed an influence on the other characteristics evaluated.

The graphical distribution of microorganisms showed that the strains *Ps. atacamensis* (SI 443), *Ps. granadensis* (SI 422), *Ps. reinekei* (SI 441) and *Ps. granadensis* (SI 450) had the best PGP performances for siderophore production, phosphate solubilization and HCN and NH_3_ production, whereas *B. tequilensis* (SI 319), *Ps. lini* (SI 287), *Br. frigoritolerans* (SI 433) and *Ps*. *frederiksbergensis* (SI 307) highlighted the best performances for IAA production and ACC deaminase activity.

## 4. Discussion

Microbial diversity is one of the main factors characterizing natural ecosystems; soil is considered one of the best storehouses of useful microorganisms in the world. Although the role of most of these microorganisms is still unknown, scientific progress is providing a better comprehension of the specific ecological functions of soil microorganisms [33]. The microbial community encountered in soil includes bacteria, molds and protozoa; some of them are free-living, while others live in symbiotic form with various species of plants. These microorganisms can be in different types of relationship with the plants, since their role can be indifferent, harmful or favorable [33].

In order to evaluate the PGP abilities of the indigenous bacteria present in an anthropogenic soil, in this work 63 soil bacteria, belonging to three different *phyla* (Actinobacteria, Firmicutes and Proteobacteria) and isolated by a modified Sicilian soil [24], were tested in vitro for their PGP abilities. The scope of this research was to establish if the natural bacterial community resident in this site was able to support plant growth. Indeed, anthropogenic soils are not cultivated soon after modification, in order to give the microbial community a certain period of time to find a new equilibrium after the addition of exogenous material. Practical observations in the area of Palma di Montechiaro (Sicily) under study suggested around five years as the optimal time before starting grape plant cultivation.

In general, Actinobacteria are one of the richest phyla of PGPB; bacteria belonging to *Frankia* genus are involved in symbiosis with plants. Other Actinobacteria, especially *Arthrobacter*, *Micrococcus* and *Streptomyces,* are considered plant growth boosters, although they do not take part in symbiotic relationships [34,35]. Firmicutes represent the most important phylum involved in PGP. In particular, *Bacillus* was thoroughly proven to exert positive effects in soil, which is directly related to plant growth [36,37,38,39,40]. These bacteria use the broadest range of PGP mechanisms, such as production of siderophores and IAA, ACC-deaminase activity and phosphate dissolution [41]. Proteobacteria are also counted as PGPB. Among these, Alphaproteobacteria include 13 different genera, especially *Ensifer* and *Cupriavidus*, that are recognized as symbiotic organisms with legumes [33]; Gammaproteobacteria include the genus *Pseudomonas* whose species might be plant pathogenic, but also PGP, especially by producing auxins, gibberellins, cytokinin, and ethylene, as well as by asymbiotic nitrogen fixation and mineral solubilization [42,43]. Some strains of *Pseudomonas aurantiaca* are also involved in HCN and siderophore production, and solubilization of phosphate [44]. Among phytohormones, one of the most important groups is undoubtedly composed of auxins. They influence many cellular functions [45]; despite the fact that a number of naturally occurring auxins have been identified, IAA has received the greatest attention by the scientific community, and the terms auxin and IAA are commonly used synonymously. In plants, IAA is typically found in conjugated forms that are primarily involved in IAA catabolism transport, storage and protection [45,46]. Tryptophan, a common precursor in root exudates, is widely converted in nature into IAA by plants and PGPB through the metabolic processes of transamination and decarboxylation [47]. It has been proposed that IAA produced by PGPB may shield cells from the harmful effects of environmental stresses [48]. Furthermore, it was demonstrated by several authors that the bacterial IAA promoted lateral and adventitious root growth, improving mineral and nutrient uptake [45]. Auxin is widely produced by soil bacteria, with an estimated 80% of soil bacteria showing this characteristic. In fact, numerous strains of soil bacteria, as well as *Alcaligenes, Azotobacter, Azospirillum, Enterobacter, Klebsiella, Pantoea, Pseudomonas*, *Rhizobium* and *Streptomyces*, have been found to express this property [45,47]. Not all isolates tested in this study produced IAA, even though the majority of them produced IAA amounts in the range of those described in previous works; for example, levels between 29 and 71 mg L^−1^ were reported by Tara and Saharan [49] for *Br. frigoritolernas* strains, and in this study, *Br. frigoritolernas* showed values between 0 and 7.37 mg L^−1^. According to Wahyudi et al. [50], 5 strains of *Streptomyces* that were obtained from soybean rhizosphere produced IAA in the range of 5.25–12.04 mg L^−1^, which are values closest to those found for our *Streptomyces* strains (*Streptomyces mauvecolor* and *Streptomyces silaceus*), with 3.15 and 5.33 mg L^−1^, respectively. Our results showed that several *Bacillus* species were able to produce IAA, although their levels were quite variable. It is among *Bacillus* genus that our investigation found the highest IAA production, and this could be explained by the high efficiency of this genus to utilize nutrients supplied by the plant through exudates [51]. Regarding *Pseudomonas*, our results were comparable to those found by some authors [51,52], while other authors reported higher values than ours [33,51,53,54]; this heterogeneity in IAA production is attributable to multiple biochemical pathways, genetic control, and environmental influences [55].

The production of ammonia is another notable aspect related to PGPB. In particular, this compound indirectly influences plant development. In this study, not all isolates were able to produce ammonia. Plants use released ammonia as a source of nutrients. Furthermore, in nitrogen-rich soils, an accumulation of ammonia can cause the soil to become alkaline; these soil conditions prevent the growth of some fungi [56,57]. According to Joseph et al. [58], a high percentage of bacteria belonging to the *Pseudomonas* genus resulted positive to ammonia tests; but with regards to the *Bacillus* genus, the results are not comparable to those found in the previous study, because only 3 out of 14 strains were positive in the NH_3_ test.

HCN production is of particular importance in soil, because its overproduction might suppress plant fungal infections [59]. Furthermore, the generation of hydrogen cyanide is positively correlated with nitrogen accumulation, root elongation, biomass production, and shoot elongation [60]. *Bacillus*, *Pseudomonas*, *Serratia*, *Arthrobacter* and *Stenotrophomonas* are considered PGPB and are involved in HCN production [61]. Although a wide variety of bacterial genera are recognized as HCN producers, in our study only eight strains, all belonging to the genus *Pseudomonas*, were found to be producers of this volatile substance.

Several proteins involved in a variety of both microbial and plant processes need iron as a cofactor. Thus, iron is essential for plant growth and development. The fourth most common element in the crust of the earth is iron [62]. Unfortunately, a relatively little amount of this element is in the ferric ion (Fe^3+^) form that is assimilated by living organisms [63]. This obstacle is overcome by several bacteria, especially siderophores, tiny organic compounds produced by microbes in iron-limited environments that increase the capability to absorb iron [64,65]. Moreover, the presence of siderophores allows plants to absorb iron despite the presence of other metals such as cadmium and nickel [66]. Producers of siderophores, in addition to chelating iron, can also adsorb other heavy metals such as lead arsenic, aluminum, magnesium, zinc, copper, cobalt and strontium [62]; for this reason, these microorganisms can be used as bioremediators. Our results demonstrate that a high number of bacteria tested were positive in the siderophores test. In particular, 31 strains belonging to nine different genera were seen to be producers of these organic compounds. Among these, four strains were *Br. frigoritolerans*; indeed, the same species resulted positive to this test in the work of Rasool et al. [67], demonstrating the good aptitude of this species as PGP. *Bacillus megaterium* was one of the best siderophore producers in our study, and similar findings were reported by Wani and Khan [68]. Also, a *Serratia* strain tested positive for this character, confirming what had already been reported by Koo and Cho [69]. Finally, *Pseudomonas* are widely utilized as bioremediators thanks to their ability to produce siderophores [70,71,72]. Our results showed that all *Pseudomonas* strains tested tested positive for siderophores, showing their important role in soil.

The production of ethylene is an important strategy developed by plants to induce a rapid protective response in reaction to external stress [73]. Basically, plant response consists of two phases: the first phase is characterized by a small peak of ethylene production, while in cases of chronic or intense stress, plants react with a huge production of ethylene that can lead to a variety of processes, including aging, chlorosis and defoliation, which impede plant growth [74]. As described by several authors, the production of ethylene in higher plants is regulated by three enzymes: S-adenosyl-L-methionine (SAM), 1-aminocyclopropane-1-carboxylic acid (ACC) and ACC oxidase [45]. Some microorganisms possess a particular enzyme (ACC deaminase) that is able to split the precursors of ethylene ACC into ammonia and α-ketobutyrate [75,76], thus reducing the amount of ethylene formed. In our work, 19 out of 63 bacteria did not show the presence of the ACC deaminase enzyme. Among those positive for this character, *Br. frigoritolerans* (the strain SI 433) showed the highest value of α-ketobutyrate with 80.58 nmol /g protein h, while the other bacteria within the *Brevibacterium* genus (ranging between 9.05 and 40.60 nmol/g protein h of α-ketobutyrate) behaved similarly to the brevibacteria isolated and screened by Tiryaki et al. [77]. In our study, strains of the *Pseudomonas* genus showed a high percentage of positivity to ACC deaminase activity. In particular, *Ps. Lini* (SI 287) showed the highest value of α-ketobutyrate among this genus (62.28 nmol/g protein h), and similar values for the same bacteria species were reported by Palacio-Rodríguez [78]. Regarding *Bacillus*, values of α-ketobutyrate synthetized by ACC deaminase of our strains were similar to those found by Misra et al. [79].

The last PGP test performed in this work was phosphate solubilization. Phosphorus, as well as nitrogen, is one of the most important elements involved in plant nutrition [80]. In particular, phosphorus has a role in every major metabolic function, including energy transmission, signal transduction, respiration, macromolecular biosynthesis and photosynthesis. Despite being one of the elements most present in soils, both in organic and inorganic form, 95–99% of the phosphate is contained in the insoluble, immobilized, and precipitated forms, making absorption by plants quite difficult. For this reason, solubilization and mineralization of phosphate is one of the most important characteristics of PGPB. These bacteria have a low rhizosphere pH thanks to their secretion of different organic acids, such as carboxylic acid and succinic acids, which causes the bound forms of phosphate like Ca_3_(PO_4_)_2_ to be released in calcareous soils [81,82]. Inorganic phosphate solubilization has also been linked to the release of H^+^ [83] and the creation of chelating agents [84,85]. Furthermore, phosphorous biofertilizers can increase the nitrogen fixation and implement the availability of substances like iron and zinc [62]. Only eight bacterial strains in our study tested positive in the phosphate solubilization test, and the genus mainly represented was *Pseudomonas*. As reported by several works, several bacteria expressing this character and known as PGP belong to this genus [82,86,87,88,89]. The biggest halo (8 mm) was observed for the strain *Ps. lini* SI 27. Zhang et al. [90] measured wider halos for the same species. *Streptomyces silaceus* SI 332 showed a solubilization halo of barely 4 mm, the smallest one registered in the screening. *Streptomyces* genus is actually active in solubilizing soil phosphate [91,92,93,94]. None of the *Bacillus* strains solubilized phosphates, although it is reported as a genus particularly active from this perspective [95,96,97,98].

Finally, all PGP traits of the bacteria tested were analyzed by multivariate statistical analysis to better individuate the strains characterized by the best PGP characteristics. For PCA, it emerged that two technological traits, i.e., IAA production and ACC deaminase activity, were positively related to each other, in contrast to those reported by Castellano-Hinojosa et al. [98], while according to the same authors, siderophore production was positively related to phosphate solubilization. The strains *Ps. Atacamensis* (SI 443), *B. Tequilensis* (SI 319), *Ps. Lini* (SI 287), *Br. frigoritolerans* (SI 433), *Ps*. *frederiksbergensis* (SI 307), *Ps. granadensis* (SI 422), *Ps. Reinekei* (SI 441) and *Ps. granadensis* (SI 450) had the largest contributions to the total variance, according to PCA analysis, so by this analysis we can say that these eight bacterial strains possessed the best PGP performances, and they could be used for single or consortium inoculations in vivo in order to test their abilities as PGPB, as reported by several works [99,100,101,102].

## 5. Conclusions

In conclusion, the PGP screening showed that all bacteria analyzed displayed positivity to at least one of the tests applied; these findings highlight that the microbial biodiversity present in the anthropogenic soil seven years after creation reached a certain capacity to provide support for plant growth functions. In addition, eight bacterial strains distributed among *Pseudomonas*, *Bacillus* and *Brevibacteria* genera were recognized as excellent producers of PGP substances. Additional research will be needed to evaluate the in vivo PGP performance of these microorganisms in fields cultivated for table grapes.

## Figures and Tables

**Figure 1 microorganisms-10-02167-f001:**
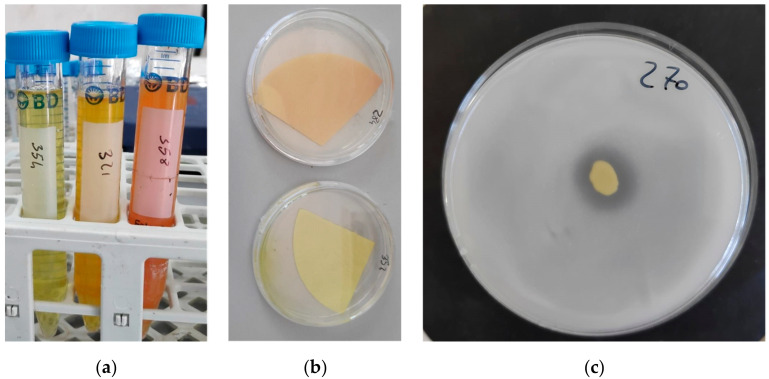
Visual results of plant-growth promoting (PGP) tests: (**a**) NH_3_ production; (**b**) HCN production; (**c**) halo generated by phosphate solubilization.

**Figure 2 microorganisms-10-02167-f002:**
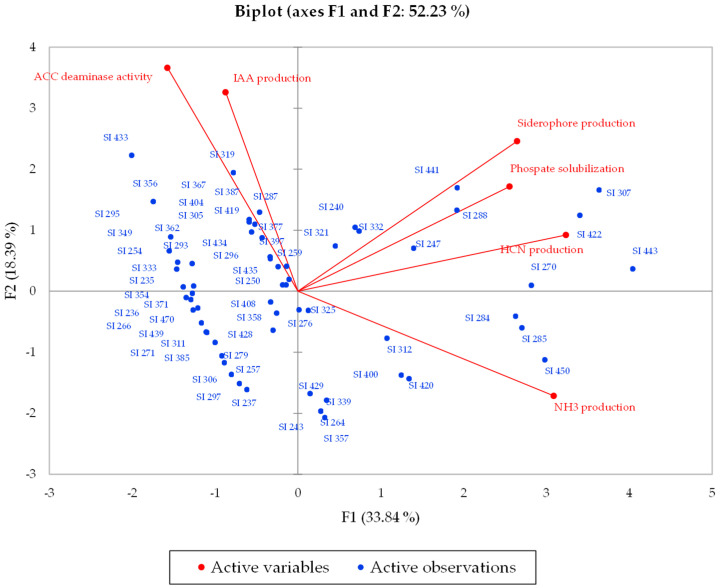
Principal component analysis (PCA) among the screened bacterial strains and the PGP traits.

**Table 1 microorganisms-10-02167-t001:** Main plant-growth promoting (PGP) traits for which the bacteria were screened.

Strains	Species	IAA Production (mg/L)	NH_3_ Production	HCN Production	ACC Deaminase Activity (nmol α-Ketobutyrate/g h)	Siderophore Production	Phospate Solubilization
SI 257	*Br. frigoritolerans*	2.50 ± 0.3 ^ghijk^	−	−	0 ^w^	−	−
SI 264	*Br. frigoritolerans*	2.50 ± 0.4 ^ghijk^	+	−	0 ^w^	−	−
SI 325	*Br. frigoritolerans*	0 ^k^	−	−	37.17 ± 2 ^hijk^	+	−
SI 312	*Br. frigoritolerans*	1.76 ± 0.2 ^ijk^	+	−	22.72 ± 5 ^mnopqr^	+	−
SI 385	*Br. frigoritolerans*	3.46 ± 0.2 ^defghij^	−	−	9.05 ± 3 ^stuvw^	−	−
SI 333	*Br. frigoritolerans*	3.75 ± 0.61 ^cdefghij^	−	−	40.60 ± 6 ^fghi^	−	−
SI 349	*Br. frigoritolerans*	6.72 ± 0.65 ^ab^	−	−	28.91 ± 6 ^jklm^	−	−
SI 400	*Br. frigoritolerans*	1.76 ± 0.1 ^ijk^	+	−	0 ^w^	+	−
SI 433	*Br. frigoritolerans*	7.37 ± 0.5 ^a^	−	−	80.58 ± 4 ^a^	−	−
SI 387	*Br. frigoritolerans*	6.72 ± 0.6 ^ab^	−	−	16.18 ± 2 ^opqrst^	+	−
SI 293	*R. erythropolis*	2.14 ± 0.2 ^hijk^	−	−	72.56 ± 4 ^ab^	−	−
SI 250	*R. equi*	3.15 ± 0.15 ^efghij^	−	−	17.52 ± 7 ^nopqrs^	+	−
SI 271	*N. globerula*	4.81 ± 0.2 ^abcdefgh^	−	−	0 ^w^	−	−
SI 279	*Str. mauvecolor*	3.15 ± 0.32 ^efghij^	−	−	4.20 ± 1 ^uvw^	−	−
SI 332	*Str. Silaceus*	5.33 ± 0.34 ^abcdefg^	−	−	0 ^w^	+	3.01
SI 362	*M. hydrocarboxydans*	5.57 ± 0.52 ^abcdef^	−	−	34.85 ± 5 ^ijkl^	−	−
SI 371	*M. oxydans*	4.83 ± 0.38 ^abcdefgh^	−	−	23.98 ± 5 ^lmnopq^	−	−
SI 295	*A. nitrophenolicus*	5.08 ± 0 ^abcdeg^	−	−	55.96 ± 6 ^cde^	−	−
SI 429	*P. aurescens*	3.46 ± 0.14 ^defghij^	+	−	0 ^w^	−	−
SI 236	*I. cucumis*	5.81 ± 0.31 ^abcde^	−	−	9.05 ± 3 ^stuvw^	−	−
SI 254	*Pb. simplex*	6.50 ± 0.37 ^abc^	−	−	20.15 ± 3 ^mnopqrs^	−	−
SI 397	*Pb. simplex*	4.03 ± 0.07 ^bcdefghij^	−	−	7.51 ± 2 ^tuvw^	+	−
SI 259	*Pb. simplex*	1.33 ± 0.3 ^jk^	−	−	49.49 ± 5 ^efg^	+	−
SI 306	*B. tequilensis*	3.15 ± 0.22 ^efghij^	−	−	0 ^w^	−	−
SI 296	*B. tequilensis*	3.46 ± 0.35 ^defghij^	−	−	25.23 ± 4 ^lmnop^	+	−
SI 319	*B. tequilensis*	6.27 ± 0.3 ^abcd^	−	−	51.66 ± 5 ^def^	+	−
SI 354	*B. tequilensis*	6.72 ± 0.1 ^ab^	−	−	0 ^w^	−	−
SI 305	*B. megaterium*	5.57 ± 0.3 ^abcdef^	−	−	27.69 ± 4 ^jklmn^	+	−
SI 404	*B. megaterium*	6.94 ± 0.88 ^ab^	−	−	7.51 ± 0.7 ^tuvw^	+	−
SI 408	*B. megaterium*	5.08 ± 0.11 ^abcdefg^	+	−	38.32 ± 6 ^ghij^	−	−
SI 470	*B. megaterium*	4.83 ± 0.04 ^abcdefgh^	−	−	14.81 ± 2 ^pqrstu^	−	−
SI 266	*B. megaterium*	6.04 ± 0.1 ^abcde^	−	−	0 ^w^	−	−
SI 339	*B. halotolerans*	0 ^k^	+	−	34.85 ± 3 ^ijkl^	−	−
SI 419	*B. halotolerans*	4.83 ± 0 ^abcdefgh^	−	−	27.69 ± 4 ^jklmn^	+	−
SI 297	*B. mohavensis*	1.33 ± 0.13 ^jk^	−	−	7.51 ± 2 ^tuvw^	−	−
SI 311	*B. cabrialensis*	4.83 ± 0.4 ^abcdefgh^	−	−	0 ^w^	−	−
SI 428	*B. cabrialesii*	6.94 ± 0.04 ^ab^	+	−	0 ^w^	−	−
SI 356	*T. saccharophilus*	6.27 ± 0.21 ^abcd^	−	−	64.36 ± 6 ^bc^	−	−
SI 243	*E. adherens*	2.50 ± 0.3 ^ghijk^	+	−	0 ^w^	−	−
SI 240	*Sn. meliloti*	5.57 ± 0.3 ^abcdef^	−	−	0 ^w^	+	2.96
SI 235	*Sn. meliloti*	6.50 ± 0.3 ^abc^	−	−	9.05 ± 1 ^stuvw^	−	−
SI 420	*S. quinivorans*	0 ^k^	+	−	17.52 ± 3 ^nopqrs^	+	−
SI 237	*C. respiraculi*	0 ^k^	−	−	18.85 ± 3 ^mnopqrs^	−	−
SI 435	*V. paradoxus*	5.33 ± 0.4 ^abcdefg^	−	−	0 ^w^	−	3.1
SI 439	*V. paradoxus*	5.33 ± 0.42 ^abcdefg^	−	−	0 ^w^	−	−
SI 321	*St. indicatrix*	6.50 ± 0.3 ^abc^	+	−	26.47 ± 4 ^klmno^	+	−
SI 358	*L. soli*	4.57 ± 0.37 ^abcdefghij^	+	−	37.17 ± 3 ^hijk^	−	−
SI 357	*L. soli*	2.14 ± 0.02 ^hijk^	+	−	0 ^w^	−	−
SI 377	*St. rhizophila*	4.57 ± 0.23 ^abcdefghi^	−	−	18.85 ± 2 ^mnopqrs^	+	−
SI 367	*Ps. Plecoglossicida*	6.50 ± 0.28 ^abc^	−	−	20.15 ± 3 ^mnopqrs^	+	−
SI 247	*Ps. brassicacearum*	3.75 ± 0.6 ^cdefghij^	−	+	16.18 ± 2 ^opqrst^	+	−
SI 307	*Ps. frederiksbergensis*	3.15 ± 0.2 ^efghij^	+	+	48.40 ± 5 ^efgh^	+	3.80
SI 441	*Ps. reinekei*	6.50 ± 1 ^abc^	+	+	45.09 ± 6.4 ^efghi^	+	−
SI 443	*Ps. atacamensis*	2.14 ± 0.2 ^ghijk^	+	+	10.54 ± 1 ^stuvw^	+	3.85
SI 422	*Ps. granadensis*	5.08 ± 1 ^abcdefg^	+	+	16.18 ± 3 ^opqrst^	+	3.2
SI 450	*Ps. granadensis*	0 ^k^	+	+	12.00 ± 1 ^rstuv^	+	−
SI 434	*Ps. moorei*	4.83 ± 0.65 ^abcdefgh^	−	−	14.81 ± 3 ^pqrstu^	+	−
SI 270	*Ps. lini*	2.83 ± 0.1 ^fghijk^	+	−	0 ^w^	+	4.99
SI 285	*Ps. lini*	2.83 ± 0.22 ^fghijk^	+	+	0 ^w^	+	−
SI 276	*Ps. lini*	2.14 ± 0.12 ^hijk^	−	−	13.42 ± 2.4 ^qrstu^	+	−
SI 284	*Ps. lini*	3.46 ± 0.2 ^defghij^	+	+	0 ^w^	+	−
SI 288	*Ps. lini*	6.27 ± 0.34 ^abcd^	+	−	16.2 ± 3 ^opqrst^	+	4
SI 287	*Ps. lini*	3.15 ± 0.1 ^efghij^	−	−	62.28 ± 4.8 ^bcd^	+	−

Abbreviations are as follows: IAA, indole acetic acid; ACC, 1-aminocyclopropane-1-carboxylate deaminase activity; *A.*, *Arthrobacter*; *B.*, *Bacillus*; *Br.*, *Brevibacterium*; *C.*, *Cupriavidus*; *E.*, *Ensifer*; *I.*, *Isoptericola*; *L.*, *Lysobacter*; *M.*, *Microbacterium*; *N.*, *Nocardia*; *P.*, *Paenarthrobacter*; *Pb.*, *Peribacillus*; *Ps.*, *Pseudomonas*; *R.*, *Rhodococcus*; *S.*, *Serratia*; *Sn.*, *Sinorhizobium*; *St.*, *Stenotrophomonas*; *Str.*, *Streptomyces*; *T.*, *Terribacillus*; *V.*, *Variovorax*. Data within a column followed by the same letter are not significantly different for *p* ≤ 0.05 according to Tukey’s test.

## Data Availability

Not applicable.

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
