# Peer review of "Plant Growth-Promoting Activities of Bacteria Isolated from an Anthropogenic Soil Located in Agrigento Province"

_microorganisms, 2022, doi:10.3390/microorganisms10112167_

Round 1
Reviewer 1 Report
In my opinion, this paper entitled “Plant Growth Promoting Activity of Bacteria Isolated from an Anthropogenic Soil” may be interesting for reader of Microorganisms Journal. Nevertheless, there are some items that should be addressed before acceptance:
From the Title of the manuscript remove an
The MS contains 21% similarity with other previous published article, mainly material and method. Reduce the material and method Plagiarism. (Plagiarism file attached)
Reduce the % of Plagiarism upto 15%
The resolution of the figures is very low, improve (Figure 2)
Line 33: About it, no need of about it, just start from In South Italy
Line 87: Mentioned the absorbance wavelength
Line 122-124: Correct Superscript and Subscript
Line 155-158 : All genus and species name will be in Italics
Line 166: Not clear
Line 169: Barely ? Check
Line 172-173: Sometime Pseudomonas written in abbreviation and sometime full name. Use abbreviation in entire manuscript. Follow 1 style in whole manuscript
The writing could be improved by strengthening the connectivity between paragraphs. There are several places where new topics are introduced and connections to the previous subject are not clear. Read whole manuscript and correct wherever required.
Introduction:
The introduction does not clearly state the purpose of the research – please amend.
Conclusions
The conclusions are too general. Please make them more specific.
Carefully read whole manuscript line by line and improve the sentence formation
All over the manuscript many different concepts are presented but not well linked each other. Many times, result difficult to understand the purpose of a sentence in the context of the paragraph.
Carefully read whole manuscript line by line and improve the sentence formation and typo errors.

Author Response
Answers to Reviewer 1:
In my opinion, this paper entitled “Plant Growth Promoting Activity of Bacteria Isolated from an Anthropogenic Soil” may be interesting for reader of Microorganisms Journal. Nevertheless, there are some items that should be addressed before acceptance.
A.U. Thanks a lot for the comments. All your requests have been considered and the changes in the text were highlighted in yellow.
From the Title of the manuscript remove an.
A.U. Thanks for your suggestion, but due to the request of reviewer 2 to put the location site of the field, the indefinite article has been maintained
The MS contains 21% similarity with other previous published article, mainly material and method. Reduce the material and method Plagiarism. (Plagiarism file attached). Reduce the % of Plagiarism up to 15%.
A.U. Thanks for your suggestion. The manuscript was modified to reduce plagiarism below 15%. In particular, the program Plagiarisms X was used to compare our manuscript to those reported in international literature. Final similarity dropped down to 11%.
The resolution of the figures is very low, improve (Figure 2)
A.U. Fig. 2 extension has been modified to enhanced metafile. The resolution improved. Probably, the problem is due to pdf conversion.
Line 33: About it, no need of about it, just start from In South Italy
A.U. Deleted as suggested. (L33).
Line 87: Mentioned the absorbance wavelength
A.U. Added as suggested (L90-91).
Line 122-124: Correct Superscript and Subscript
A.U. Corrected (L128).
Line 155-158: All genus and species name will be in Italics
A.U. Modified (L211-215).
Line 166: Not clear
A.U. Modified for clarity (L174).
Line 169: Barely? Check
A.U. The term barely was replaced by “only” (L176).
Line 172-173: Sometime ”Pseudomonas” written in abbreviation and sometime full name. Use abbreviation in entire manuscript. Follow 1 style in whole manuscript
A.U. Thanks for the clarification. Microbiological orthographic rules for nomenclature binary system established that genus names are reported in the extended form at first citation (abstract is section apart), then, only if the same species is encountered the abbreviation is used for genus. However, genus abbreviations have been harmonized throughout the text.
The writing could be improved by strengthening the connectivity between paragraphs. There are several places where new topics are introduced and connections to the previous subject are not clear. Read whole manuscript and correct wherever required.
The introduction does not clearly state the purpose of the research – please amend.
A.U. Thanks for your comment. The introduction was better developed to better introduce the readers to the main aim of the work (L37-39, 65-70).
The conclusions are too general. Please make them more specific. Carefully read whole manuscript line by line and improve the sentence formation All over the manuscript many different concepts are presented but not well linked each other. Many times, result difficult to understand the purpose of a sentence in the context of the paragraph. Carefully read whole manuscript line by line and improve the sentence formation and typo errors.
A.U. We reread the manuscript and modified all section according to your suggestions (L 28,29, 133-142, 220-222, 226-227, 229-235, 236, 239, 240-241, 247, 281-282, 298299, 377-380)
Reviewer 2 Report
The manuscript addressed isolation of plant growth promoting bacteria from an anthropogenic soil, and determination of their biostimulating traits. Some comments were made as can be seen in the attached file. However, the main concern was the lack of data that show the growth enhancement of economically important plants. Such experiments are highly recommended before acceptance of the manuscript. I suggest selecting the highly competitive strains in terms of their PGP traits and assess their plant growth promotion on selected crops. Inoculation can be done either singly or as a consortium.
Additionally, rationale for conducting this study should be clearly reported.

Author Response
Answers to Reviewer 2:
The manuscript addressed isolation of plant growth promoting bacteria from an anthropogenic soil, and determination of their biostimulating traits. Some comments were made as can be seen in the attached file. However, the main concern was the lack of data that show the growth enhancement of economically important plants. Such experiments are highly recommended before acceptance of the manuscript. I suggest selecting the highly competitive strains in terms of their PGP traits and assess their plant growth promotion on selected crops. Inoculation can be done either singly or as a consortium.
Additionally, rationale for conducting this study should be clearly reported.
A.U. Thanks for your comments. With regards to the comments reported directly in the manuscript, all the comments have been addressed and changes highlighted in green.
The aim of this preliminary work was to establish if, after 7 years from its creation, the anthropogenic soil object of study hosted a microbial community able to generically support the growth of plants. To this purpose, in vitro tests to evaluate PGP traits were performed on bacteria isolated from this very soil. To date, we did not carry out any in-vivo tests, because of the high specificity of the soil. In fact, the anthropogenic soils within the area under investigation are generally created to improve the profitability in terms of grape table vineyards. Thus, any in-vivo test should be conducted on grape plants that require a long period of cultivation to obtain repeatable results. Of course we are very grateful to the reviewer for this suggestion. Please consider also that the research project is longer than that described in this work, but for this preliminary screening we chosen the “microorganisms” journal because it gives more value to aspects related to microbiota than plants. Once again, thanks a lot for your suggestion, maybe the aim of the work was not clear in the previous version. For this reason, also to encounter reviewer 1 requests, Abstract, Introduction and Conclusions were modified to better explain the work aims (L382-383).
